# Starch Structure of Raw Materials with Different Amylose Contents and the Brewing Quality Characteristics of Korean Rice Beer

**DOI:** 10.3390/foods12132544

**Published:** 2023-06-29

**Authors:** Jiyoung Park, Hye Young Park, Hyun-Jung Chung, Sea-Kwan Oh

**Affiliations:** 1Department of Central Area Crop Science, National Institute of Crop Science (NICS), Rural Development Administration (RDA), Suwon 16429, Republic of Korea; pjy2812@korea.kr (J.P.); phy0316@korea.kr (H.Y.P.); 2Division of Food and Nutrition, Chonnam National University, Gwangju 61186, Republic of Korea; 3National Institute of Crop Science (NICS), Rural Development Administration (RDA), Chuncheon 24219, Republic of Korea

**Keywords:** rice beer, beer aroma, beer quality, starch structure

## Abstract

This study aimed to explore suitable processing materials for rice beer (RB) production by analyzing the starch structure of the raw materials utilized for brewing beer and the quality characteristics of RB. We used malt, employing the Heugho cultivar as the main ingredient, and produced beer containing 30% rice. The regular amylose-containing cultivars Samgwang (SA) and Hangaru (HA) and the high-amylose-containing cultivar Dodamssal (DO) were used as adjuncts. Distribution of the short molecular chains of the starch amylopectin was the highest for SA and malt at 29.3% and 27.1%, respectively. Glucose content was the highest in the wort prepared with 100% malt and 30% SA + 70% malt. The alcohol content in SA RB and HA RB was higher than that in beer prepared with 100% malt. DO RB had the least bitterness and volatile components, such as acetaldehyde and ethyl acetate. The three rice cultivars tested in this study are suitable as starch adjuncts for RB production. The characteristics of RBs varied depending on the molecular structure of the ingredients, irrespective of their amylose contents. SA could be considered a craft beer with quality characteristics and rich flavor components, similar to 100% malt beer, compared to other RBs.

## 1. Introduction

Beer is one of the most popular traditional alcoholic beverages in Korea and the rest of the world. Nutritionally, beer is abundant in amino acids, carbohydrates, minerals, and phenolic compounds, the contents of which depend on the raw ingredients and the brewing processes, including steeping and fermentation [1].

In South Korea, small-scale breweries that use native ingredients aim to promote the beer industry through quality differentiation and enhancement. The Liquor Tax Act in Korea was revised in January 2018 to promote the use of domestic ingredients and commercialize the craft beer industry. The revised law lowered the tax rate from 72% to 30% based on the shipment volume for beer prepared utilizing at least 20% rice. This has stimulated considerable interest in the development of rice beer (RB) among regional craft beer brewers [2].

The main ingredients for brewing beer, including hops, yeast, and barley malt, are mostly imported into South Korea, and the proportion of domestic barley ingredients is only 5% [3]. Beers are typically made from barley malt; however, other unrefined grains, such as corn, rice, rye, oats, and wheat, called adjuncts, are also used. These ingredients are employed as a source of fermentable material. Nevertheless, studies have explored the potential of these adjuncts as the main ingredient, which not only imparts various benefits but can also reduce the cost of beer production [4].

Starch determines the quality of grains; however, the effects of starch structure during processing have not been precisely determined. The starch content of barley is lower than that of other grains, at approximately 55–60%. Moreover, differences have also been reported in the amylose content and structure of barley and other grains. However, barley varieties used for brewing beer show differences in the distribution of amylose chain lengths but not in amylose content. Furthermore, differences in starch structure are not observed when measuring the fermentation efficiency in the hydrolysis of starch to fermentable sugar [5]. Fermentable sugars can be supplemented by adjuncts, which usually have no enzymatic activity, indicating that starch hydrolysis mostly depends on malt enzymes. This indicates that starch and its structure, including amylose and amylopectin content and the amylopectin chain length, can affect the product quality of beer and depend on the grains included in the brewing process [6].

Rice is a staple food for Koreans, but there has been a gradual decline in annual rice consumption of approximately 1–2 kg per year, reaching 61.9 kg in 2016, because of various social changes, including the westernization of food culture and an increase in dual-income couples [7]. Moreover, 49,000 tons of rice are mandatorily imported each year under the Free Trade Agreement, and the opening of rice imports in 2015 has led to the additional importation of various types of rice for processing. Furthermore, domestic rice production has increased because of incentives to develop high-yield varieties. The combined results have led to plentiful rice stocks and difficulty in adjusting supply and demand. This has necessitated continuous efforts to increase domestic rice consumption and create new sources of demand.

Globally, rice (*Oryza sativa* L.) is one of the most important adjuncts for brewing beer; however, not all rice varieties are suitable [6]. In the beer industry, rice malt is mostly used as an unrefined adjunct alongside barley malt because of its lower saccharogenic power. Rice malt improves extraction during saccharification [8,9].

Although some Korean studies have focused on the manufacture of beer using rice and barley, almost no research has focused on the quality characteristics of beer and wort depending on the starch structure of raw ingredients.

Heugho barley, which has a black aleurone layer containing polyphenols and anthocyanins, was developed in 2014 and has attracted considerable interest from the industry as a source material for beer manufacturing because of the increased consumer demand for distinguished products with health functionality. Samgwang (SA) was developed as cooked rice in 2003 and has become popular because of its excellent taste. Hangaru (HA) is characterized by white rice grains resembling those of glutinous rice [10]. HA is easily pulverized into a powder form, making it widely utilized for producing rice flour and processing. Dodamssal (DO) contains resistant starch and is known for its beneficial action against insulin resistance [11]. In this study, we analyzed the starch structures of malt produced from newly developed Korean black barley and three cultivars of rice used as adjuncts. We also explored the quality characteristics of RB made from these ingredients.

## 2. Materials and Methods

### 2.1. Materials

Heugho, a two-row barley (*Hordeum vulgare* L.) cultivar developed for beer production in Korea, was utilized in this study. Malting was performed following the procedure described by Kim et al. [12] with some modifications. With the use of a malting machine at the National Institute of Crop Science, 10 kg malting was performed by steeping, germination, and kilning. Undamaged barley seeds that did not pass through a 2.5 mm standard sieve were selected and washed. The steeping procedure was performed at 18 °C for 50 h to achieve 42% steeping. Subsequently, germination (at 17 °C for 100 h) and kilning (at 45 °C for 14 h and 75 °C for 7 h) were performed. Finally, curing was performed at 80 °C before termination. The dried malt was cooled to ≤35 °C, and the roots were removed for subsequent composition analysis and starch extraction.

Three rice cultivars, SA (intermediate-amylose content), HA (intermediate-amylose content), and DO (high-amylose content), were used as adjuncts in this study.

Malt and rice were ground using a grinder (CT293 CyclotecTM, FOSS Analytical Co. Ltd., Suzhou, China), and the flour was filtered through a 100-mesh standard sieve (No. 100 ASTM E11, Standard Test Sieve Scientific Co., Ltd., Wonju, Republic of Korea) and used for analyses. The starch was isolated using the alkaline steeping method, and after drying at 40 °C, the starch was ground in the same way as malt and rice flours.

### 2.2. Production of RB

RB was brewed utilizing a method described in a previous study with modifications [2]. Malt and rice were roughly crushed using a malt grinder and employed for brewing. Five kilograms of raw material in a ratio of 70% malt to 30% rice (*v*/*v*) were mixed with 20 L of brewing water and heated. The rice was added first, boiled for 30 min at 95 °C, and cooled to facilitate the saccharification of unprocessed rice. The malt was added when the water temperature reached 65 °C. Saccharification was performed from 72 °C to 78 °C for 1 h to obtain the wort. Subsequently, the wort, which consisted of the residue from both malt and rice, was removed, and hops were added (2.5%, 50 g, *w*/*v*) in two portions: once at the beginning of boiling at 100 °C and again 10 min before the end of boiling at 100 °C for 60 min. The boiled wort was then cooled and transferred to a fermentation vessel. Dry yeast for top fermentation was added at a rate of 0.58% (11.5 g, *w*/*v*) based on the volume of the wort, and the first round of fermentation was conducted at 20 °C for 7 days. Subsequently, the beer was transferred to a 20 L keg and subjected to maturation at 2 °C in a cold storage room for 3 days until the quality of the beer was evaluated. The beer in the control group was made only of malt and was produced following the same procedure using only malt without adding rice.

### 2.3. Composition Analysis of Malt and Rice Flours

The moisture, crude ash, crude protein, and crude fat content in malt and rice flours were analyzed following the methods described by Park et al. [13]. The moisture content was measured after incubation at 105 °C for 2 h. The crude ash content was determined after incubation at 600 °C for 5 h. The crude protein was measured using the semi-micro-Kjeldahl method (Kjeltec 2400 AUT; Foss Tecator, Hilleroed, Denmark). The crude fat content was determined following the Soxhlet extraction method using a Soxtec System HT 1043 extraction system (Foss Tecator). The total starch and amylose contents were determined utilizing a total starch assay kit and an amylose/amylopectin assay kit in the Megazyme kit (Megazyme International, Ltd., Wicklow, Ireland), respectively.

### 2.4. Microscopic Observation of Malt and Rice Starch

Malt and rice starch samples were gold-coated to increase their conductivity, and their particle sizes were evaluated under a scanning electron microscope (SEM-3000; Hitachi Ltd., Tokyo, Japan) at 1000× and 2000× magnifications.

### 2.5. X-ray Diffractometry

The crystalline structures of the starches were analyzed using an X-ray diffractometer (XRD, PANalytical, X’pert MPD high-resolution XRD; Almelo, The Netherlands). The diffractometer was operated at 40 kV and 40 mA with a scanning range of 5–40° (2θ) and a scan rate of 2.0 min. The starch crystallinity was quantitatively calculated according to the method described by Lopez-Rubio et al. [14] using peak-fitting software (Origin 6.0, Microcal, Northampton, MA, USA).

### 2.6. Molecular Weight Distribution

The molecular weights of barley and malt starch were analyzed using high-performance size-exclusion chromatography (HPSEC) performed on an HPSEC system (HELLOS, Wyatt Technology Co., Santa Barbara, CA, USA) and fitted with a multi-angle laser light scattering (MALLS) and refractive index (RI) detector. The starch samples (1.0 g), supplemented with 1–2 drops of ethanol, were prepared in 100 mL 90% dimethyl sulfoxide (DMSO) for 1 h at 100 °C, as described in a previous study with minor modifications [15]. The solubilized starch molecules were precipitated by adding ethanol (500 mL) and then centrifuged and dried. Solubilized starch (0.5%) was prepared in a 0.1 M NaOH buffer by heating the solution for 30 min at 95 °C. The solution was then neutralized with HCl (pH 7), autoclaved for 10 min, and filtered using a 0.45-µm syringe filter. The prepared sample was injected into an HPSEC analyzer fitted with an RI (Waters 2414, Waters Co., Milford, MA, USA) and a MALLS detector. Separation was achieved using SEC columns (TSK G5000 PW, 7.5 mm × 600, TosoBiosep, Montgomeryville, PA, USA). The mobile phase included 0.15 M NaNO_3_ and 0.02% NAN_3_ (HPLC grade, flow rate: 0.4 mL/min).

### 2.7. Chain-Length Distribution of Amylopectin

The amylopectin chain-length distributions of the barley and malt starches were determined using a previously described method [16]. The sample (10 mg) was dispersed in 2 mL 90% DMSO and boiled with continuous stirring for 20 min. The dispersion was mixed with 6 mL absolute ethanol and centrifuged at 2000× *g* for 12 min. The precipitate was dissolved in 2 mL 50 mM sodium acetate buffer (pH 3.5) and heated in a boiling water bath with continuous stirring for 20 min. After the solution was equilibrated to 37 °C, isoamylase (E-ISAMY, 4.16 µL, 240 U/mg, Megazyme Co., Wicklow, Ireland) was added, and the starch solution was incubated at 37 °C with constant stirring at 150 rpm for 24 h. The samples were boiled for 10 min to inactivate the enzymes. An aliquot (200 μL) of debranched starch was diluted with 2 mL NaOH (150 mM). The sample was filtered using a 0.45-µm nylon syringe filter and was analyzed using high-performance anion-exchange chromatography coupled with pulsed amperometric detection (HPAEC–PAD) on a Dionex ICS-5000 system (Thermo Fisher Scientific Inc., Waltham, MA, USA) fitted with a CarboPac PA200 column (3 × 250 mm). Separation was achieved using a gradient eluent with 150 mM NaOH and 0.6 M sodium acetate in 150 mM NaOH at a flow rate of 0.5 mL/min.

### 2.8. Composition Analysis of Monosaccharides and Oligosaccharides

The monosaccharide content of the wort prepared using Korean malt and the three rice varieties was determined. The sample (4 mg) was stirred without heat treatment for 30 min in 2 mL distilled water. After stirring, the supernatant was centrifuged (1600× g, 10 min, 4 °C) to achieve separation. The separated supernatant was filtered using a 0.45-μm micro filter and injected (10.0 μL) into the HPAEC-PAD on a Dionex ICS-5000 system fitted with a Carbopac PA 1 (4.0 × 250 mm). The mobile phase was 150 mM NaOH (isocratic) at a flow rate of 1.0 mL/min.

The oligosaccharide content of the wort prepared using Korean malt and the three rice varieties was analyzed in the same manner as the monosaccharide content. Separation was achieved using a gradient eluent with 500 mM NaOAc in 100 mM NaOH and 100 mM NaOH at a flow rate of 1.0 mL/min.

### 2.9. Quality Analysis of RB

The brewing characteristics of the RB were analyzed according to Analytica-EBC, a standard analysis method issued by the European Brewery Convention (EBC) as follows [17].

#### 2.9.1. Apparent Extract

After measuring the specific gravity (SG_A20/20°C_) of the decarbonized sample, the apparent extract (*E_A_*) was calculated using Equation (1):*E_A_* = −460.234 + 662.649 × SG_A20/20°C_ − 202.414 × SG_A20/20°C_(1)

#### 2.9.2. pH

The sample was transferred to a flask, sealed, and shaken to remove the gas. The mixture was then passed through a membrane filter. The pH was measured using a pH meter while stirring.

#### 2.9.3. Color

The absorbance (*A*_430_) at 430 nm was measured after diluting (dilution factor, F) the filtrate obtained by passing the sample through a membrane filter. Thereafter, the color (EBC units) was estimated using Equation (2):Color (EBC units) = *A*_430_ × 25 × F(2)

#### 2.9.4. Alcohol

One hundred milliliters of the sample were distilled, and the indicator was read on the hydrometer by adjusting it to 15 °C on the Gay-Lussac table. The alcohol content was expressed as %(*v*/*v*).

#### 2.9.5. Real Extract

After diluting the original volume with distilled water, the real extract from the de-alcoholized sample was measured using a saccharometer.

#### 2.9.6. Original extract

The original extract content was calculated using Equation (3):(3)$$\mathrm{Original}\;\mathrm{extract}\;[\%\;w/w]=\frac{\left(A\times 2.0665+E_{R}\right)\times 100}{A\times 1.0665+100}$$
where *A* represents the alcohol content, and *E_R_* represents real extract content.

#### 2.9.7. Real Degree of Fermentation (RDF)

RDF was calculated using the alcohol (*A*) and real extract (*E_R_*) contents as follows:(4)$$\mathrm{RDF}\;[\%]=\frac{2.0665\times A}{2.0665\;\times A+E_{R}}\times 100$$

#### 2.9.8. Apparent Degree of Fermentation (ADF)

Fifteen grams of yeast was added to 200 mL of the sample, which was fermented for 24 h at 20 °C. Afterward, the sample was decarbonated, and the specific gravity was measured. Subsequently, the ADF was calculated using specific gravity (*E_A.final_*), and the content (ƿ) was obtained using the following equation:(5)$$\mathrm{ADF}\;[\%]=\frac{(ƿ-E_{A.final)}\times 100}{ƿ}$$

#### 2.9.9. Bitter Taste

The sample was prepared by adding 0.5 mL of hydrochloric acid and 20 mL of isooctane to 10 mL of the decarbonized sample and shaking it at 130 ± 5 rpm for 15 min at 20 °C. The sample was centrifuged for 3 min at 1600× *g*, and the absorbance (*A*_275_) of the isooctane layer was measured at 275 nm. The bitter taste (BU) was evaluated using the following equation:BU = 50 × *A*_275_(6)

#### 2.9.10. Calcium

The analyte was prepared by mixing 20 mL of the decarbonized sample with 100 mL distilled water, 3 mL 5 M NaOH, and 0.5 mL calcein indicator. The calcein indicator was prepared by dissolving 0.2 g calcein in distilled water containing 1 mL of 5 M NaOH. The prepared analyte was titrated with EDTA until it turned orange-brown, and the calcium (mg/L) content was calculated using the following equation:Ca (mg/L) = EDTA appropriate mL × 20(7)

#### 2.9.11. Chlorine and Lactic Acid

Chlorine and lactic acid were analyzed using ion chromatography. The samples were degassed and diluted with distilled water for analysis. The guard column was AG 17C, and the analytical column was 17C; a Diones EGC 500 KOH cartridge was utilized. The flow rate was 2.0 mL/min, and the detector conductivity was 3 or 30 µS. The sample size was 50 µL.

#### 2.9.12. Volatile Compounds

The acetaldehyde, methyl acetate, normal propanol, isobutanol, isoamyl alcohol, and isoamyl acetate produced during alcoholic fermentation were analyzed using gas chromatography (Perkin Elmer Sigma 8500, Waltham, MA, USA). A Chrompack 7773 (50 m × 0.32 mm × 1.11 µm) capillary column was utilized to separate the volatile aroma components, and N_2_ was used as the carrier gas. The column temperature was maintained at 75 °C for 6 min, increased to 110 °C at 25 °C/min, and maintained for 3 min. The column temperature was then increased to 250 °C for analysis.

### 2.10. Statistical Analysis

All data are presented as the mean of triplicate measurements and were analyzed using SAS v. 9.4 (SAS Institute Inc., Cary, NC, USA). Statistical significance was determined using a one-way analysis of variance and Duncan’s multiple comparison test. *p*-values at <0.05 were considered significant.

## 3. Results and Discussion

### 3.1. Malt and Rice Flour Composition

Table 1 shows the results of the composition of the Korean malt and the three rice varieties used for manufacturing RB. The water content ranged from 6.8% to 15.6%. The DO and HA varieties, which are soft-type rice with a white outer appearance, had higher water content than the transparent SA variety. Malt had the lowest water content owing to drying during the malting process. The crude ash, crude fat, and crude protein content were the highest in malt. During brewing, barley cultivars with a protein content of 8–12% are usually selected, and a very high or low protein content can hinder the brewing process [18]. For instance, a very low protein content leads to insufficient enzymes for suitable fermentation and affects foam formation [19]. The protein content analyzed after processing the malt of the Heugho barley used in this experiment was 9.9%, which is a suitable value for brewing beer. In previous studies, it was difficult to make beer using 100% rice including rice malt, and one of the reasons was that the protein composition was different, and the low protein content of rice affected beer quality [20]. Further, in our study, the total starch content in malt (63.4%) was significantly lower than that in the rice varieties (82.3–84.7%). The amylose content was the highest in DO, followed by malt, HA, and SA. The starch content of rice was higher than that of malt, which is consistent with the results of a previous study that analyzed the chemical compositions of barley and malt [21]. The apparent amylose contents of SA and HA have been reported to be 18.2% and 19.3%, respectively [10,22]. However, in this study, they were slightly lower at 15.8% and 17.6%, respectively. The difference could be attributed to the differences in the estimation method. Park et al. reported that the content long-chain amylose was overestimated using the iodine staining method, which may explain why we obtained slightly lower values using the enzymatic starch hydrolysis method [23]. In previous studies, the amylose content of DO was reported to be 40%, but in this study, it was 35.1%, showing consistent results. However, further studies are required to understand the effects of amylose content on the initial molecular structure, wort saccharification, and wort quality.

### 3.2. Morphology of the Starch Isolated from Korean Malt and Rice Varieties

The SEM images of the starch isolated from Korean malt and the three rice varieties used to make RB at 1000× and 2000× magnifications are shown in Figure 1. The shape of the barley starch varied from oval to round, and the average size was 10 µm or larger. Starch granules with traces of enzymatic hydrolysis were observed in the malt samples (Figure 1a,b). Because unprocessed white rice was used, the surface of the particles was smooth with no signs of hydrolysis. The starches from all three rice varieties were smaller (2–10 µm) than malt starch. SA and HA were present as small polygonal particles resembling typical rice starch. Won et al. [10] reported that HA exhibited spherical starch particles when sectioned. In contrast, we observed polygonal particles similar to those in common rice, which may be attributable to modifications caused by alkali due to the use of NaOH for extraction. Unlike SA and HA, DO contained several starch grains larger than 10 µm and round starches in agglomerated form. This is consistent with the results reported in a previous study [23].

### 3.3. Crystalline Structure of the Starches

The results of the crystallinity XRD analysis of starch isolated from Korean malt and the three rice varieties and their relative crystallinity are shown in Figure 2a and Table 2, respectively. Except for DO, the malt and rice samples exhibited two non-separated peaks at 15°2Θ and 17°2Θ, two peaks at 18°2Θ, and a sharp peak at 23°2Θ, indicating a typical type A starch structure. DO showed type C crystalline characteristics, with a mixture of both type A and type B starches. The relative crystallinity was the highest in the order HA > SA > Malt > DO (Table 2).

The relative crystallinities of the crystalline and amorphous regions were measured as a proportion of the X-ray diffractograms. The relative crystallinity of barley starch is diverse at 10.71–43.21%; it is not correlated with the amylose content and decreases during germination [24]. The A-type XRD pattern of rice showed little difference in relative crystallinity depending on the amylose content, but the C-type high-amylose rice variety showed significantly lower relative crystallinity. The crystallinity is reduced by enzymatic hydrolysis; however, the crystalline structure remains unchanged [23]. The malt used in this study showed lower amylose content than the A-type varieties SA and HA and slightly higher amylose content than the C-type variety DO but showed normal relative crystallinity, and the A-type crystal pattern was thought to be unchanged. These results are consistent with those in a previous study [25].

### 3.4. Molecular Weight and Chain-Length Distribution of the Starches

Peaks 1 and 2 represent the amylopectin and amylose contents, respectively (Figure 2b). The molecular weight of amylopectin ranged from 114.5 ± 1.7 to 119.8 ± 3.2 (×10^6^ g/mol) with no difference across samples (Table 2). The mass fraction of the amylopectin peak was the highest in the order of SA > HA > malt > DO, whereas the mass fraction of the amylose peak showed the opposite order. The peak 2 ratio of DO was particularly high and was characterized by a significantly low molecular weight, consistent with a high apparent amylose content (Table 1 and Table 2).

With respect to amylopectin chain length, the proportion of short chains of degree of polymerization (DP) 6–12 was the highest in the order of SG > malt > HG > DD, with DO having a significantly lower proportion (15.3%) than the other samples (23.3–29.3%). For medium chains, DP 13–24 and 25–36, malt and SA showed the lowest proportions. The proportion of long chains of DP ≥ 37 and the average chain length was in the order DO > malt > SA > HA, all of which were significant, except for SA and HA. DO, a high-amylose variety, had the highest chain length.

The molecular weight of peak 2 of malt and DO, which have a relatively large starch particle size and generally round shape, tended to be significantly low, and the molecular weight of DO was the lowest. However, because a difference exists in the molecular weight of SA and HA with similar starch granules, we believe that various other factors such as amylose content affect the molecular weight rather than starch particles. Sun et al. [26] reported that the starch granule size affects the fine structure of amylopectin. In this study, the molecular weight of amylopectin was not significantly different in all samples, but the average chain length was high in malt and DO, showing similar results (Figure 1 and Table 2).

According to Park et al. [23], when the amylose content of rice starch is high, or following enzymatic hydrolysis, the relative amount of long amylopectin chains increases, leading to an increased mean chain length. The chain-length (molecular weight) distribution and amylose content of different varieties of barley and malt starch have been reported to be similar but different from those of rice [27]. In our study, the starch structure of malt, including the molecular weight, amylopectin chain-length distribution, and relative crystallinity, showed characteristics of normal rice and high-amylose rice, suggesting that, rather than intrinsic differences between rice and barley, the differences are caused by factors such as amylose content and starch changes during germination.

### 3.5. Monosaccharide and Oligosaccharide Composition of Wort

Table 3 shows the monosaccharide and oligosaccharide contents of wort prepared using Korean malt and the three rice varieties. The monosaccharide composition was the same for all samples, in the order of maltose > glucose > sucrose > fructose. SA wort had the highest maltose content, but this was not statistically significant. The only significant difference was observed in the glucose content (fermentable sugar) in malt and SA worts. Oligosaccharides were detected up to DP 3–7. The content of DP 3, a maltotriose, tended to be in the order malt > SA > DO > SA. DP 6, a long-chain oligosaccharide, was the most abundant in HA, whereas DP 7 was the most abundant in DO and HA, with a significant difference. DO, which had the longest average chain length, showed a high content of long oligosaccharides when processed into worts, whereas malt and SA worts, which had a relatively high percentage of short chains, showed a low content of long oligosaccharides.

According to Yu et al. [22], the addition of a starch to barley malt, irrespective of the microscopic molecular structure of the starch, significantly alters the fermentable sugar content of the wort, especially maltose; however, the effects were modified by the characteristics of barley malt, such as beta-amylase activity [27]. Because glucose is the first sugar used by yeast in the wort, a higher glucose/maltose ratio has usually been reported to produce a higher ethanol content. The addition of rice significantly reduces the glucose/maltose ratio and increases the maltotriose ratio [28]. Amylase (especially alpha-amylase) in wort prefers to hydrolyze linear starch chains, producing maltodextrins such as maltotriose rather than glucose [29]. In our study, similar to previous ones [28], beer made from barley malt showed the highest glucose content. SA showed the highest ratio of short amylopectin chains (DP 6–12) in its starch, indicating that it produced a significantly higher glucose content in the wort with added rice. Meanwhile, the high-amylose variety DO had the longest mean chain length and the highest ratio of long chains (DP ≥ 37), which resulted in a high maltodextrin content in the DP 6–7 range (Table 2 and Table 3). Because we manufactured beer at a pilot scale using only a single variety of malt, the differences in the production of fermentable sugars were determined by the molecular characteristics of rice starch and not by beta-amylase in the malt, suggesting that it is essential to select the appropriate variety of rice.

### 3.6. Quality Characteristics of RB

Using Korean malt and three rice varieties, we manufactured four types of beer (malt beer [MB] and RB containing SA, HA, or DO) with either 100% malt or 70% malt + 30% rice. The quality characteristics of the four types of beer are shown in Table 4. The apparent extract, real extract, alcohol, and original extract showed similar trends in the order of SA > HA > Malt > DO; the alcohol content was 6.6–7.7%. The pH and EBC values (color) were higher in MB (pH, 4.5; EBC, 19.1) than those in the three RBs (pH, 4.2 in all three; EBC, 13.9 [SA], 10.9 [HA], and 9.6 [DO]). Furthermore, the RDF and ADF were significantly higher in DO RB than those in the other samples. BU was the highest for MB and was the lowest for DO RB. Calcium, chlorine, and lactic acid levels were the highest in the MB and differed significantly from their contents in RBs. The propanol, isobutanol, and isoamyl alcohol levels were the lowest in MB. RB prepared with medium-amylose rice varieties (SA and HA) had the highest ethyl acetate content.

A previous study showed that the higher the EBC value, the darker the beer; the lower the EBC value, the lighter the beer [30]. This explains the reason for the darker color of MB than that of RB and the lighter color of DO RB made from high-amylose-containing rice. In a previous study [20], beer made of rice malt had a low Mailliard content because of its low soluble nitrogen content in the wort, which has a nearly pale color. In addition, the color of the variety with a low pH of wort was pale [20]. In this study, the pH of the beer with rice was lower, leading to a pale color. The pH of the RB was the lowest for the DO variety. The alcohol content ranged from 4.5 to 5.3% (*v*/*v*) in the industrial beer samples; however, the difference was large at 4.4 to 9.7% (*v*/*v*) in the craft beer with a higher alcohol content. Even in the case of bitterness, industrial beer has 8–20 IBU, whereas craft beer and beer have 6–62 IBU [31]. The results of this study are consistent with those of previous studies, showing an appropriate IBU range for craft beers. Other studies have shown that a high proportion of glucose to maltose produces high ethanol content because glucose is the first sugar fermented by yeast [32]. However, in this study, the alcohol content of SA among RBs with a high maltose content in the wort was significantly higher than that of MB with the highest glucose content and ratio, showing conflicting results. This indicates that the content of maltose is as important as that of glucose, a fermented sugar.

No difference in maltose content was found; however, the alcohol content of DO was significantly lower among all samples, including HA and DO, which had low glucose content; therefore, it is thought to be related to various factors such as the long chain length of starch and the high maltoheptaose content of wort in DO. Rice is a starch adjunct in beer brewing that provides a neutral taste, balanced flavor, and light color [9]. The main factor controlling the volatile flavor components of beer is the characteristics of the original malt. In contrast, the addition of starch affects only the content of these flavor components [28]. Propanol provides the alcohol content and sweet aroma to beer; thus, craft beer is stronger than industrial beer [33]. In this study, RB showed stronger characteristics than MB; thus, its characteristics were thought to be closer to those of handmade beer. Isoamyl alcohol provides the flavors of banana, wine, and alcohol [34]. In this study, the isoamyl alcohol content in RB was also higher than that in MB, wherein SA RB showed the highest propanol and isoamyl alcohol content, indicating that it had a rich flavor.

In DO, the contents of acetaldehyde, which represent green leaves, paint, and a fruity flavor, and ethyl acetate, which represent solvent, fruity flavor, and sweetness, were the lowest, whereas the isobutanol content was the highest. As such, the flavor characteristics of high-amylose rice differ from those of other rice varieties.

## 4. Conclusions

We performed malting and beer processing using three rice varieties with different amylose content (SA, HA, and DO) as adjuncts and Heugho barley. We analyzed the composition of the raw materials, molecular structure of the starch, composition of the wort, and quality characteristics of the beer. After analyzing the beer characteristics of each variety by adding 30% rice as a starch adjunct, the content of fermented sugars (glucose and maltose) in SA (with intermediate amylose content) was found to be high. The alcohol, propanol, and isoamyl alcohol contents were high in the three RBs, indicating that they were flavored and suitable for fermentation. DO showed a type C crystal structure and a long amylopectin chain length. It also had the lightest color, lowest alcohol content, and high ADF and RDF. Overall, the findings suggest that the three rice cultivars tested in this study are suitable for RB production as a starch adjunct; however, in the craft beer market, which requires more suitable and diverse flavors, it is important to choose suitable rice varieties; thus, further studies conducting in-depth analyses on this aspect are needed.

## Figures and Tables

**Figure 1 foods-12-02544-f001:**
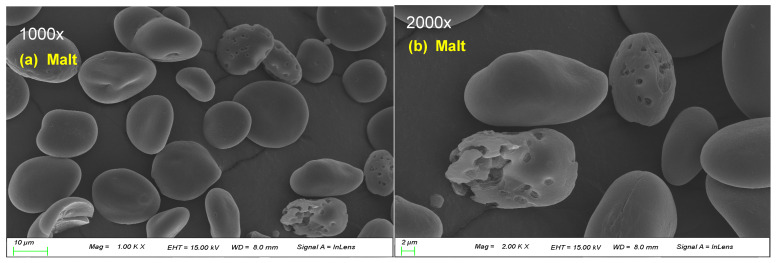
Scanning electron micrographs of starch isolated from the Korean malt and three rice varieties at 1000× (**a**,**c**,**e**,**g**) and 2000× magnification (**b**,**d**,**f**,**h**). Malt, Heugho malt; SA, Samgwang rice; HA, Hangaru rice; DO, Dodamssal rice.

**Figure 2 foods-12-02544-f002:**
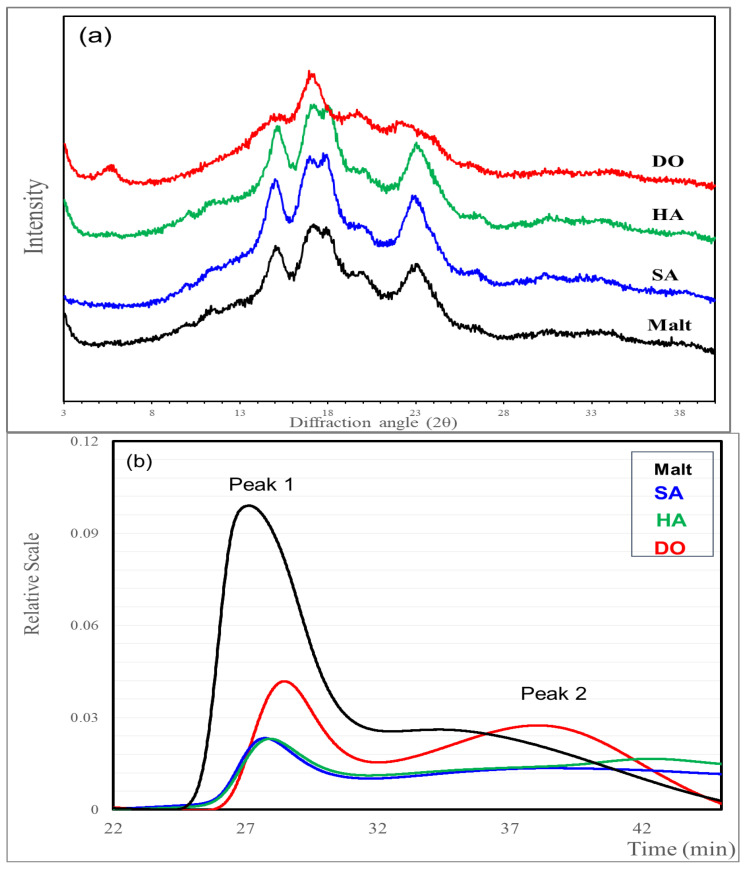
Crystallinity and chromatograms of starch isolated from Korean malt and three rice varieties. (**a**) X-ray diffraction patterns of malt and rice starches from Korean cultivars; (**b**) High-performance size exclusion chromatography chromatogram of malt and rice starches from Korean cultivars; Peak 1 (amylopectin), Peak 2 (amylose). Malt, Heugho malt; SA, Samgwang rice; HA; Hangaru rice; DO, Dodamssal rice.

**Table 1 foods-12-02544-t001:** Composition and amylose content of flours in Korean malt and three rice varieties.

Samples	Moisture	Crude Ash	Crude Fat	Crude Protein	Total Starch	Amylose
Malt	6.8 ± 0.1 ^c^	1.5 ± 0.0 ^a^	2.1 ± 0.1 ^a^	9.9 ± 0.1 ^a^	63.4 ± 0.2 ^c^	25.1 ± 0.4 ^b^
SA	13.9 ± 0.2 ^b^	0.8 ± 0.0 ^b^	1.4 ± 0.1 ^b^	5.1 ± 0.1 ^c^	84.4 ± 0.4 ^a^	15.8 ± 0.3 ^d^
HA	15.5 ± 0.3 ^a^	0.7 ± 0.0 ^b^	1.6 ± 0.1 ^b^	5.0 ± 0.1 ^c^	84.7 ± 0.3 ^a^	17.6 ± 0.6 ^c^
DO	15.6 ± 0.1 ^a^	0.7 ± 0.0 ^b^	1.5 ± 0.1 ^b^	6.2 ± 0.1 ^b^	82.3 ± 0.3 ^b^	35.1 ± 0.5 ^a^

^a–d^ Values with different letters within a column are significantly different (*p* < 0.05), as determined by Duncan’s multiple-range test. Values are presented as the mean ± SD of three independent trials. Malt, Heugho malt; SA, Samgwang rice; HA, Hangaru rice; and DO, Dodamssal rice.

**Table 2 foods-12-02544-t002:** Relative crystallinity, molecular weight, and amylopectin chain-length distribution of isolated starches from Korean malt and three rice varieties.

Sample	Relative Crystallinity (%)	Peak I		Peak II		Amylopectin Chain-Length Distribution (%)				Average Chain Length
		Mw (×10^6^ g/mol)	Mass Fraction (%)	Mw (×10^6^ g/mol)	Mass Fraction (%)	DP 6–12	DP 13–24	DP 25–36	DP ≥ 37	
Malt	25.9 ± 0.6 ^b^	117.6 ± 3.8 ^a^	63.7 ± 0.7 ^c^	12.7 ± 1.1 ^c^	36.3 ± 0.7 ^b^	27.14 ± 0.0 ^a^	48.1 ± 0.1 ^b^	13.9 ± 0.5 ^a^	10.9 ± 0.7 ^b^	20.8 ± 0.2 ^b^
SA	32.4 ± 1.3 ^a^	114.5 ± 1.7 ^a^	68.3 ± 0.6 ^a^	42.1 ± 0.5 ^a^	31.7 ± 0.5 ^d^	29.3 ± 0.4 ^a^	51.4 ± 0.1 ^a^	11.8 ± 0.2 ^b^	7.61 ± 0.2 ^c^	18.5 ± 0.4 ^c^
HA	32.8 ± 0.2 ^a^	119.8 ± 3.2 ^a^	65.6 ± 0.6 ^b^	21.3 ± 0.6 ^b^	34.4 ± 0.4 ^c^	23.3 ± 0.1 ^a^	51.4 ± 1.3 ^a^	13.7 ± 0.0 ^a^	6.2 ± 0.6 ^c^	18.6 ± 0.4 ^c^
DO	23.2 ± 0.7 ^c^	115.7 ± 5.8 ^a^	37.2 ± 0.5 ^d^	5.2 ± 0.1 ^d^	62.8 ± 0.5 ^a^	15.3 ± 0.0 ^b^	50.2 ± 0.3 ^a^	13.3 ± 0.3 ^a^	21.2 ± 0.6 ^a^	25.5 ± 0.3 ^a^

^a–d^ Values with different letters within a column are significantly different (*p* < 0.05), as determined by Duncan’s multiple-range test. Values are presented as the means ± SD of three independent trials. Malt, Heugho malt; SA, Samgwang rice; HA, Hangaru rice; DO, Dodamssal rice; DP, degree of polymerization.

**Table 3 foods-12-02544-t003:** Sugar composition of worts from Korean malt and three rice varieties.

Analysis	Standard	100% Malt	30% Rice + 70% Malt		
		Malt	SA	HA	DO
Free sugar (mg/mL)	Glucose	11.57 ± 1.02 ^a^	10.94 ± 0.81 ^a^	8.88 ± 0.23 ^b^	8.94 ± 0.13 ^b^
	Fructose	2.22 ± 0.23 ^a^	1.66 ± 0.10 ^b^	1.28 ± 0.00 ^c^	1.36 ± 0.03 ^bc^
	Sucrose	6.74 ± 0.57 ^a^	6.75 ± 0.54 ^a^	5.17 ± 0.20 ^b^	5.61 ± 0.03 ^ab^
	Maltose	63.55 ± 10.82 ^a^	80.07 ± 10.91 ^a^	65.35 ± 6.90 ^a^	65.66 ± 4.82 ^a^
Oligosaccharide (mg/mL)	Maltotriose (DP 3)	13.88 ± 0.61 ^a^	13.86 ± 0.11 ^a^	12.59 ± 0.74 ^a^	13.39 ± 0.44 ^a^
	Maltotetraose (DP 4)	3.72 ± 0.19 ^ab^	4.07 ± 0.37 ^a^	3.42 ± 0.22 ^ab^	3.34 ± 0.16 ^b^
	Maltopentaose (DP 5)	0.90 ± 0.05 ^a^	1.00 ± 0.10 ^a^	0.92 ± 0.06 ^a^	0.88 ± 0.05 ^a^
	Maltohexaose (DP 6)	0.81 ± 0.06 ^b^	0.87 ± 0.08 ^ab^	1.01 ± 0.07 ^a^	0.97 ± 0.04 ^ab^
	Maltoheptaose (DP 7)	0.32 ± 0.01 ^b^	0.31 ± 0.04 ^b^	0.44 ± 0.03 ^a^	0.49 ± 0.03 ^a^

^a–c^ Values with different alphabetic letters within a column are significantly different (*p* < 0.05), as determined by Duncan’s multiple-range test. Values are presented as the means ± SD of three independent trials. Malt, Heugho malt, SA Samgwang rice, HA Hangaru rice; DO, Dodamssal rice. No other monosaccharides (such as rhamnose, mannose, ribose, or cellobiose) were detected.

**Table 4 foods-12-02544-t004:** Characteristics of beer prepared using Korean malt and three rice varieties.

Analysis	(Unit)	100% Malt Beer (MB)	30% Rice Beer (RB)		
		Malt	SA	HA	DO
Apparent extract	(^o^P)	3.0 ± 0.1 ^b^	3.5 ± 0.2 ^a^	3.3 ± 0.5 ^ab^	2.3 ± 0.2 ^c^
pH		4.5 ± 0 ^a^	4.2 ± 0.1 ^b^	4.2 ± 0.1 ^b^	4.2 ± 0.1 ^b^
Color	(EBC)	19.1 ± 0.6 ^a^	13.9 ± 0.4 ^b^	10.9 ± 0.2 ^c^	9.6 ± 0.1 ^d^
Alcohol	(*v*/*v*%)	7.1 ± 0.3 ^bc^	7.7 ± 0.0 ^a^	7.3 ± 0.3 ^ab^	6.6 ± 0.6 ^c^
Real extract	(^o^P)	5.4 ± 0.2 ^b^	6.2 ± 0.2 ^a^	5.8 ± 0.5 ^ab^	4.7 ± 0.3 ^c^
Original extract	(^o^P)	15.8 ± 0.5 ^b^	17.4 ± 0.1 ^a^	16.5 ± 0.7 ^ab^	14.5 ± 1.1 ^c^
Real Degree of Fermentation (RDF)	(%)	67.7 ± 0.3 ^b^	66.9 ± 0.5 ^b^	67.1 ± 1.1 ^b^	69.9 ± 0.5 ^a^
Apparent Degree of Fermentation (ADF)	(%)	81.5 ± 0.3 ^b^	80 ± 0.7 ^b^	80.5 ± 1.6 ^b^	84.4 ± 0.5 ^a^
Bitter taste	(BU)	27.7 ± 2 ^a^	24.6 ± 2 ^ab^	22.3 ± 0.3 ^bc^	18.4 ± 3.6 ^c^
Calcium	(mg/L)	28.5 ± 2 ^a^	23.6 ± 1.4 ^ab^	19.3 ± 0.8 ^c^	22.5 ± 1.3 ^b^
Chlorine	(mg/L)	152.9 ± 0.9 ^a^	143.4 ± 1.6 ^b^	141.6 ± 5.4 ^c^	123 ± 9.6 ^d^
Lactic acid	(mg/L)	295.4 ± 5.7 ^a^	273.2 ± 16.1 ^b^	255.7 ± 1.4 ^c^	252.4 ± 5.8 ^c^
Acetaldehyde	(mg/L)	2.7 ± 0.2 ^a^	2.6 ± 0.8 ^a^	3 ± 0.3 ^a^	2.2 ± 0.4 ^a^
Ethyl acetate	(mg/L)	16.7 ± 1 ^b^	20.5 ± 0.4 ^a^	21.4 ± 1.3 ^a^	14.7 ± 0.5 ^c^
Normal propanol	(mg/L)	36.1 ± 1.2 ^c^	43.6 ± 0.7 ^a^	42.7 ± 1.3 ^ab^	40.5 ± 2.5 ^b^
Isobutanol	(mg/L)	41.6 ± 1.6 ^c^	55.2 ± 3.5 ^b^	54.5 ± 2.5 ^b^	65.3 ± 4.5 ^a^
Isoamyl alcohol	(mg/L)	95.1 ± 1.9 ^b^	103 ± 1 ^a^	96.5 ± 2.3 ^b^	97.8 ± 4.5 ^b^
isoamyl acetate	(mg/L)	0.9 ± 0.1 ^b^	1 ± 0.1 ^a^	1.1 ± 0.0 ^a^	0.7 ± 0.1 ^c^

^a–d^ Values with different alphabetic letters within a column are significantly different (*p* < 0.05), as determined by Duncan’s multiple-range test. Values are presented as the means ± SD of three independent trials. Malt, Heugho malt; SA Samgwang rice; HA Hangaru rice; and DO Dodamssal rice.

## Data Availability

The data presented in this study are available on request from the corresponding author.

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
