# Peer review of "Starch Structure of Raw Materials with Different Amylose Contents and the Brewing Quality Characteristics of Korean Rice Beer"

_foods, 2023, doi:10.3390/foods12132544_

Round 1

Reviewer 1 Report

In this study, the authors analyzed the starch structures of malt produced from a Korean black barley and three cultivars of rice used as adjuncts. The quality characteristics of the resultant rice beer were also investigated.This paper will be of interest to the readership of Foods. However, there are still some questions need to be further discussed.

1.The role of starch during processing has been emphasized.The effect of protein should be mentioned. 

2.The starch structures of malt were analyzed. However, to reveal the changes during processing, it is recommended to investigate the starch structure in brewer's spent grain.

3.It is recommended to use SI unit. Change M to mol/L.

Line 118 change “five kilograms” to “5 kg”

Line 122 change “one h” to “1 h”

...

Author Response

Reviewer 1

Comments and Suggestions for Authors

In this study, the authors analyzed the starch structures of malt produced from a Korean black barley and three cultivars of rice used as adjuncts. The quality characteristics of the resultant rice beer were also investigated.This paper will be of interest to the readership of Foods. However, there are still some questions need to be further discussed.

→ We appreciate your review of our manuscript. We have revised the manuscript according to the comments received from the reviewers and have sought professional help for English language editing. We have responded to each of the comments as follows.

1.The role of starch during processing has been emphasized. The effect of protein should be mentioned. 

→ We agree with your opinion. Protein content is important when choosing the type of malt, which is the main ingredient of beer. This study was based on starch rather than protein and the aim was to understand the selection of rice raw materials, which include one type of malt with starch adjuncts. In the introduction, we decided that it was unsuitable to mention proteins. According to your suggestion, we have added protein-related references and a discussion to section 3.1 (Red font).

  1. The starch structures of malt were analyzed. However, to reveal the changes during processing, it is recommended to investigate the starch structure in brewer's spent grain.

→ Thank you for your suggestion. We used malt and rice as raw materials to make beer. First, after the saccharification process, the wort was prepared, and beer was produced through wort fermentation. We analyzed the quality of all raw materials used for processing and the final products. Therefore, to reveal changes in the process, we believe that the quality of malt and rice starch analyzed in this study, the monosaccharides and oligosaccharides in the wort, and beer quality, including alcohol content, are sufficient and appropriate. However, we think it is necessary to study the spent grain during brewing, as you suggested, in the future for understanding raw material yield and economic feasibility.

3.It is recommended to use SI unit. Change M to mol/L.

→ Thank you for pointing this out. We have revised Table 2 accordingly.

Comments on the Quality of English Language

Line 118 change “five kilograms” to “5 kg”

→ Thank you for suggestion. But our English editor commented that “Ideally, sentences should not begin with numerals in academic writing. “

*Five kilograms of raw material in a ratio of 70% malt to 30% rice (v/v) was mixed with 20 L of brewing water and heated.  

Line 122 change “one h” to “1 h”

→ Thank you for pointing this out. We have revised it accordingly.

Reviewer 2 Report

This manuscript is about starch structure of raw material with different amylose contents and brewing quality characteristics of Korean rice beer. It is interesting and I think minor revision of manuscript should be made. You can find my comments in below:

1. The manuscript must be revised grammatically and the English level of it must be improved by a native editor.

2. The authors must re-write the abstract and conclusion sections. Some sentences written in abstract must go to conclusion part and some in conclusion must be written in abstract section.

3. In introduction part, it is better to decrease the number of sentences about the export and import of ingredients of beer and add some other information that could be more valuable.  

4. In lines 116 to 130, please add more details about the beer production procedure and give an appropriate reference.

5. In lines 271 to 289, please more compare the obtained results with other same researches.

6. In result and discussion part, please discuss about the effect of different parameters and experiments on each other. For example, what is the relationship between the results obtained from the SEM images and the molecular weight results and do they confirm each other or not.

7. Please increase the DPI value and quality of figures especially SEM images.

The manuscript must be revised grammatically and the English level of it must be improved by a native editor.

Author Response

Reviewer 2

Comments and Suggestions for Authors

This manuscript is about starch structure of raw material with different amylose contents and brewing quality characteristics of Korean rice beer. It is interesting and I think minor revision of manuscript should be made. You can find my comments in below:

→ We appreciate your review of our manuscript. We have revised the manuscript according to the comments received from the reviewers and have sought professional help for English language editing. We have responded to each of the comments as follows:

  1. The manuscript must be revised grammatically and the English level of it must be improved by a native editor.

→ The manuscript has been reviewed by a native English editor, who has proofread the text for language and grammar.

  1. The authors must re-write the abstract and conclusion sections. Some sentences written in abstract must go to conclusion part and some in conclusion must be written in abstract section.

→ Thank you for pointing this out. In general, when the same sentences are used in the abstract and the conclusion, it may lead to repetition, and it is often pointed out that these sections should be more succinct. Furthermore, the abstract and the conclusion sections are similar, but the content within the sections differs. We have paraphrased the sentences considering these comments and organized them according to each section. However, in line with your comments, we have revised some parts of the abstract and conclusions so that important information can be retained.

  1. In introduction part, it is better to decrease the number of sentences about the export and import of ingredients of beer and add some other information that could be more valuable.  

→ We have deleted a few sentences as follows to improve the readability of the text according to your suggestion.

Line 35-41: Beer consumption in South Korea increased continuously until 2018. Despite a slight plateau owing to the COVID-19 pandemic, the reported beer output in 2019 was still high at 50.8% of the total alcoholic beverage output. Recently, the import volume of foreign beer has markedly increased by 6.8-fold from 2012 to 2019, reflecting its popularity[2]. Addi-tionally, the craft beer market (small-scale beer breweries producing 5–75 kL beer per batch) grew to 40 million dollars in 2017, demonstrating a 2-fold increase compared with 2016.

  1. In lines 116 to 130, please add more details about the beer production procedure and give an appropriate reference.

→ We have described the methods and references for brewing rice beer in Section 2.2. Some words and sentences have been revised after reviewing the section.

  1. In lines 271 to 289, please more compare the obtained results with other same researches.

→ Thank you for your suggestion. In accordance with the opinion of another reviewer that starch was emphasized in Section 3.1, we have added a description about protein (Line 273-280).

  1. In result and discussion part, please discuss about the effect of different parameters and experiments on each other. For example, what is the relationship between the results obtained from the SEM images and the molecular weight results and do they confirm each other or not.

→ We have added the result and the related discussion according to your suggestion ( L361-369)

  1. Please increase the DPI value and quality of figures especially SEM images.

→ We have replaced the SEM image with a picture of the raw data according to your comment. The changed figure clearly shows the starch granules.

Comments on the Quality of English Language

The manuscript must be revised grammatically and the English level of it must be improved by a native editor.

→ The manuscript was reviewed for grammar and language by a native English speaker.
